# Correlation between Pathogenic Determinants Associated with Clinically Isolated Non-Typhoidal *Salmonella*

**DOI:** 10.3390/pathogens10010074

**Published:** 2021-01-15

**Authors:** Boimpoundi Eunice Flavie Ouali, Tsyr-Huei Chiou, Jenn-Wei Chen, I-Chu Lin, Ching-Chuan Liu, Yu-Chung Chiang, Tzong-Shiann Ho, Hao-Ven Wang

**Affiliations:** 1Department of Life Sciences, National Cheng Kung University, Tainan 701, Taiwan; l58047022@mail.ncku.edu.tw (B.E.F.O.); thchiou@mail.ncku.edu.tw (T.-H.C.); ichulin89@gmail.com (I-C.L.); 2Department of Microbiology and Immunology, College of Medicine, National Cheng Kung University, Tainan 701, Taiwan; jc923@mail.ncku.edu.tw; 3Center of Infectious Disease and Signaling Research, National Cheng Kung University, Tainan 701, Taiwan; liucc@mail.ncku.edu.tw; 4Department of Pediatrics, National Cheng Kung University Hospital, College of Medicine, National Cheng Kung University, Tainan 701, Taiwan; 5Department of Biological Sciences, National Sun Yat-sen University, Kaohsiung 80424, Taiwan; 6Department of Biomedical Science and Environment Biology, Kaohsiung Medical University, Kaohsiung 80708, Taiwan; 7Center for Bioscience and Biotechnology, National Cheng Kung University, Tainan 701, Taiwan; 8Marine Biology and Cetacean Research Center, National Cheng Kung University, Tainan 701, Taiwan

**Keywords:** non-typhoidal *Salmonella*, virulence factor, pathogenicity, clinical isolates

## Abstract

Non-typhoidal and Typhoidal *Salmonella* are bacterial pathogens source of worldwide and major disease burden. Virulent determinants of specific serovars belonging to non-typhoidal *Salmonella* have been extensively studied in different models, yet the pathogenesis of this group of bacteria and the development of clinical symptoms globally remains underexplored. Herein, we implemented microbiological and molecular procedures to investigate isolate virulence traits and molecular diversity, likely in association with disease severity. Our results show that selected clinical isolates from a tertiary referring hospital, depending on the richness of the environment and isolate serotypes, exhibited different, and sometimes controversial, virulence properties. The tested strains were susceptible to Ceftriaxone (90%) with decreasing reactivity to Trimethoprim–Sulfamethoxazole (72%), Chloramphenicol (64%), Ampicillin (48%), Gentamicin (44%), and Ciprofloxacin (2%). Disc susceptibility results partially correlated with minimum inhibitory concentration (MIC); however, special attention must be given to antimicrobial treatment, as a rise in multi-resistant isolates to Trimethoprim–Sulfamethoxazole (2/38 µg/mL), Minocycline (8 µg/mL) and Ampicillin (16 µg/mL) has been noticed, with two isolates resistant to Ceftazidime (16 µg/mL). By comparison to previous molecular epidemiology studies, the variation in the gene profiles of endemic pathogens supports the need for continuous and up-to-date microbiological and molecular reports.

## 1. Introduction

*Salmonella* is a food-borne pathogen notorious for causing various symptomatic events in humans, from self-limiting pathologies to more complicated forms of infection. The infectious pathogen origin of enteric fever or acute gastroenteritis includes the non-typhoidal strains, the typhoidal strains, and the invasive non-typhoidal strains recently found in Sub-Saharan Africa [1,2,3]. Non-typhoidal strains responsible for food poisoning are widespread globally. The World Health Organization estimates infections due to non-typhoidal *Salmonella* enterica to bear the highest burden of foodborne diseases, with 180 million diarrheal cases, and Africa and South East Asia were identified as the largest exposed regions [4,5]. Diseases and outbreaks are mostly attributed to contaminated vegetables, meats, drinks, and animal exposure [6,7,8,9]. In Taiwan, non-typhoidal *Salmonella* is still a prevalent pathogen [10] which causes the highest annual incidence of bacteremia among the population in the South [11].

In recent years, salmonellosis management has improved considerably due to multidrug resistance trace-back methods, real-time surveillance, and the monitoring of outbreaks [10,12]. However, despite this progress, the prevalence of salmonellosis remains globally consistent and has even increased in some areas [5]. The expansion of industrialization, the evolution of microbial host adaptation, and the emergence of antibiotic-resistant clones largely contribute to the persistence of this disease [13].

The immune condition of the human host, in addition to bacterial serotype and virulence, plays a clinically important and concomitant role in the severity of the diseases caused by *Salmonella* [14]. The pathogen utilizes a large variety of virulence factors, including plasmid virulence genes, cell surface structure, flagella proteins, and pathogenicity islands (SPIs) to trigger disease by invading the host and escaping host defenses [15,16]. *Salmonella* is recognized to bind to host cell targets, to be engulfed, and to migrate through the host blood or lymphatic circulatory systems, resulting in bacteremia, septicemia, more complicated forms of the disease [17,18,19] and long-lasting carriage [20]. To invade a host cell, *Salmonella* injects invasion signaling molecules through the type III secretion system encoded by the SPI-1 which, in turn, activate the host cell signaling pathways that lead to the internalization of the pathogen and, subsequently, the reconstruction of the cytoskeleton actin protein [19,21]. The SPIs carry the genes responsible for invasion, survival, and extra-intestinal spreading [21]. Hence, bacteria can be restricted to the gastrointestinal tract or can be expanded to other organs through the bloodstream.

Serotyping, genotyping, and pulsed-field gel electrophoresis have been extensively used for disease surveillance in Taiwan; however, microbiological examination of the bacterial physiology, coupled with molecular epidemiology studies, is neglected. Herein, we seek to examine the correlation between virulence determinants in association with *Salmonella* pathogenicity by investigating the distribution of virulence traits among *Salmonella* clinical isolates and by comprehending their interconnection.

## 2. Results

### 2.1. Overview of Salmonella Isolates Adherence and Invasiveness to Epithelial Cells

To determine the ability of the isolates to attach to and pervade epithelial cells, FHs 74 Int cells were infected with the bacterial inoculum. While no statistical differences were noticed for isolates invading the cells, most of the blood and stool isolates adhered onto and invaded the epithelial cells (Figure 1). The blood isolate B2 and the stool isolates S4 and S5 exhibited the highest adhesion rates, whereas B2, followed by S4, had the greatest invasion rates among the isolates. B2 showed a significant increase in adhesion rate compared to B1. The adhesion and invasion rates of B2, respectively, approximate 20% and 12%, whereas S4 and S5 had each an adhesion rate of about 15%.

### 2.2. Environmental Conditions Affecting Bacterial Growth

To analyze the bacterial growth status and its possible relation with infection, *Salmonella* isolates were grown in a lysogeny broth and minimal medium, with the absorbance measured at 600 nm. The blood (Figure 2a) and stool isolates (Figure 2b) showed a high potency to multiply in the LB medium with a continuous exponential phase. On the other hand, in the M9 culture media, the growth of the blood (Figure 2c) and stool isolates (Figure 2d) decreased relatively, with a noticeable short log phase and bacteria reaching the stationary phase after 9 h of incubation. A drastic reduction in B2 isolate multiplication was noted in the depleted M9 media.

### 2.3. Motility of the Bacterial Isolates

Bacterial motility was measured to assess the ability of the isolates to spontaneously and actively move and to differentially swarm and swim on a semisolid surface. The 12 clinical isolates were all capable of swarming (Figure 3a) and swimming (Figure 3b) on 0.5% and 0.3% LB-agar, respectively. Partly based on the colony pattern formation described by Kearns [22], swarmer isolates exhibited dendritic, bull’s-eye, or a thick circular colony pattern, whereas swimmer isolates essentially presented a uniformly round, featureless colony. The dispersion of the blood isolates on swarm plates was not statistically different (Figure 3c), whilst stool isolate S5 had the significantly highest swarming dispersion area by comparison to S4 (Figure 3d). No significant differences were supported for the swimming dispersion surface of blood (Figure 3e) and stool (Figure 3f) isolates. However, we noticed that the stool S1 had a lesser trend to swim and swarm on agar plates.

### 2.4. The Effects of the Intrinsic Nature of the Bacteria and Culture Medium on Biofilm Formation

Most of the 12 bacterial isolates were not able to organize into a biofilm. The blood isolates B2 solely exhibited a substantial ability to form a relatively weak cell aggregate when cultured in the LB medium. In M9 medium, isolates B1, B3, S3, and S6 were identified as weak biofilm producers, whereas the B2 isolate was recognized as a strong biofilm producer. The absorbance of B2 cultured in M9 was equal to 0.97 against 0.1 in LB (Figure 4a) and resulted in a more pronounced purple color in the corresponding wells (Figure 4b). However, the differences in biofilm formation levels among the isolates were not statistically supported.

### 2.5. Salmonella Isolates Genotyping by Multiplex PCR

Three independent multiplex PCR assays were performed for each group of genes for the purpose of virulence gene profiling. A comparison to the 100 bp Plus DNA ladder indicated that the observed bands of *InvA*, *SpvC*, and *FliC* (Appendix A) with *HilA*, *FimH*, and *SopB* (Appendix A) were consistent throughout in terms of the estimated length of each amplicon. The targeted genes *InvA*, *FimH*, and *HilA* were all found in the 12 clinical isolates tested. Out of the bacteria, 50% had both *SpvC* and *SopB* but lacked *FliC*; 33.33% had *SopB* and *FliC* but lacked *SpvC*; *FliC*, *SpvC* and *SopB* were all absent in the products generated by the isolate S5, while all the amplicon bands were found in the electrophoresed product issued from isolate S1 (Table 1).

Products from multiplex PCR were run on agarose gels. The presence of the specific gene was attested by the visualization of a band at their corresponding size.

### 2.6. Comparative Virulence Gene Expression and Serotyping

To quantify the expression of virulence genes, quantitative PCR was performed using total RNA extracted from the 50 bacteria isolates. All the isolates expressed *InvA*, *FimH*, and *SopB* mRNA, with values between 0.67 and 4.96, 0.94 and 5.62, 0.26 and 1.16 arbitrary units, respectively (a.u.) (Figure 5). Overall, 30% of the isolates expressed *FliC* and 34% expressed *SpvC* at relatively low values. Out of the 50 isolates, only one blood isolate did not express *HilA*. Respective positive and negative controls with *S.* Typhimurium and *E.* coli were used for comparison purposes and results’ validation. Serotyping of the 50 isolates based on the O-antigen showed that the most prevalent serogroups were B and D1, each with 30% (15/50), followed by the serogroup C2 (20%, 10/50), and the serogroups C1 and E, which identically counted for 10% of the isolates (5/50).

### 2.7. Salmonella Isolates In Vitro Susceptibility to Antibiotics

About 90% of the 50 isolates showed sensitivity to Ceftriaxone. Decreased sensitivity was noticed with Trimethoprim–Sulfamethoxazole, Chloramphenicol, Ampicillin, Gentamicin, and Ciprofloxacin (Table 2). Among the isolates, 50% and 36% were greatly resistant to Ampicillin and Chloramphenicol, respectively, whereas Ciprofloxacin and Gentamicin had intermediate effect on, respectively, 76% and 48% of the bacteria tested.

All the isolates were sensitive to Cefepime at the minimum inhibitory concentration (MIC) breakpoint value of ≤1 and ≤0.25 µg/mL for Carbapenems (Ertapenem, Imipenem, and Meropenem). Although Ampicillin, Trimethoprim-Sulfamethoxazole, and Chloramphenicol were not considered to be good empirical regimens for *Salmonella* infections due to the high frequency of resistance [23], we found that most of the stool isolates were still susceptible to Trimethoprim–Sulfamethoxazole.

## 3. Discussion

The microbiological and molecular characterization of virulence determinants associated with human pathogens is primordial for effective disease control and management. A series of tests were performed here on *Salmonella* clinical isolates to examine their virulence factor profiles and the correlation between them and the bacterial pathogenicity. Initially, microbiological analysis was conducted on 12 *Salmonella* clinical isolates which included adherence and invasion assay, growth, motility, biofilm formation, and genotyping assays. For consideration of a more representative sample size and for wider bacterial screening, the number of isolates was eventually increased to a total of 50. The 50 isolates were then considered for the quantification of virulence gene expressions and antimicrobial testing. The 12 clinical isolates were able to attach and invade small intestine cells regardless of their isolation origins. This suggests that the isolation sites within the host do not necessarily define *Salmonella* virulence. In an early study, Suez and colleagues indicated that invasive non-typhoidal *Salmonella* generally does not present a high cell invasion competency in vitro [24]. In parallel, very few studies have been conducted using FHs-74 Int as a model for pathogenic *Salmonella* infection. FHs 74 Int, infected by a mammalian intestinal protozoan parasite, revealed that the cell is more susceptible to pathogen invasion and better mimics an in vivo infection compared to Caco-2 epithelial cells [25]. Although not discussed in the present paper, this was also confirmed with data in our preliminary studies with Caco-2 cells. Thus, our study proposes the relevant use of FHs 74 Int cells for *Salmonella* clinical isolate infection investigations.

The blood and stool isolates exhibited similar growth rates which, seemingly, were affected by environmental nutrient availability. However, it was interesting to find that fast-growing isolates such as B5 and B6 did not present a high potency to adhere/invade epithelial cells, suggesting no direct correlation between the isolate growth rate and their degree of invasiveness. Concurrently, it is known that bacteria that multiply may also undergo space colonization for nutrient uptake and escaping macrophages and drugs within the host [26,27]. Thereby, the fast-growing isolates might exploit this ability to trigger other responses, which do not include host cell invasion. The tested blood and stool isolates showed an ability to move on a semi-solid surface, and various *Salmonella* strains were also found to be motile [28,29]. Bacteria helical fimbriae and flagellum are known to drive swimming and swarming motility, respectively [30]. Nonetheless, the high swarmer isolates S5, as well as some blood and stool isolates, were shown to not express *FliC,* suggesting that other genes such as *FljB* might take part in the formation of flagella in the absence of *FliC* in those specific isolates [31]. This hypothesis has, however, not been verified in the present work, but constitutes a topic for further studies. Regarding the motile isolates which do not invade the cells, we assumed that flagella recruitment might not direct invasiveness, but may contribute to it, since the adherent strain S5 was highly motile. Our results suggested that the isolates are highly competitive in terms of nutrient availability in specific microbiota. While Liu et al. suggested intact motility to be an invasion-related factor for the specific case of *S.* Typhi [32], we observed that B2 and S4, with high invasion rates, did not exhibit a particularly high motility potency. This implies either an alteration in the intrinsic motility of the isolates or a species-dependent characteristic.

*Salmonella* flagella and motility play a concomitant role in biofilm initiation by generating the starting signal on a suitable surface [33]. Our data showed that the sole isolate identified as a strong biofilm producer was also the most invasive. This finding implies a correlation between bacterial invasiveness and biofilm formation potency, as stated earlier [34]. However, selection pressures acting on biofilm including host adaptation may inhibit invasive serovars’ ability to form a biofilm for the pathogen to improve its survival [34,35,36,37]. This may explain why the isolates S4 and S5, which lost the biofilm formation potency, were still invasive. The biofilm formed by the strain B2 was also influenced by nutrient availability, since salt variation in the environment affected *Salmonella* motility and persistence within the host [33,38]. The lysogenic medium LB provides all the nutrients required for the isolates to grow, whereas the minimal medium triggered stress and survival responses, resulting in biofilm formation. Biofilm is also known as an important virulence factor used by bacteria to escape antibiotic treatment [33,39]. We then hypothesized that the blood isolate B2 might display distinct features in reaction to antibiotic treatments. The MIC results showed that the invasive and biofilm-forming isolate was, in fact, resistant to a wide range of antibiotics. To prevent overgeneralization, all the isolates were submitted to the exact same conditions, and their phenotypes were compared between them. Thus, reference strain controls were not used in the present case for the microbiological investigations.

The ability of *Salmonella* to grow, to be motile, and to form a biofilm was directed by the presence of virulence genes. Genes encoding for effector proteins and transcription regulation were targeted to depict their contribution to *Salmonella* pathogenicity. Effector proteins, including SopB, FimH, InvA, and FliC are involved in epithelial cell adhesion, invasion, and colonization [21,26,40,41], while transcriptional regulator HilA, and SpvC are shown to positively regulate virulence-associated genes and invasion genes [40,42]. From an early epidemiological study, most *Salmonella* clinical isolates from hospitals in Taiwan were shown to harbor the *SpvC* virulence plasmid [43]. While this was partly the case in our investigation, the clinical implication has yet not been elucidated. We speculated that the lack of *SpvC* in B2, S4, and S5 might contribute to increased adherence to FHs 74 Int cells. Moreover, bacteria with no *SpvC* gene exhibited either an ability to form a biofilm, to grow quickly, or to invade epithelial cells, conditions that lead to symptomatic manifestations in a human host. Nevertheless, these outcomes, drawn from the study of 12 isolates, may not be sustained when considering all 50 isolates, which show a more diverse virulence gene expression profile. Most of the isolates from serogroup B, C1, C2, and E do not express *SpvC* mRNA whereas most isolates from serogroup D1 lack *FliC* mRNA (Appendix A). A more recent study stated that the *Salmonella* serogroup C2 was more involved in host bacteremia [44].

The emergence of multidrug-resistant *Salmonella* strains has led investigators to make attempts to find an appropriate, effective antibiotic treatment against invasive serotypes involved in salmonellosis [45,46]. Our disc diffusion results indicated that Ceftriaxone is an effective antibiotic against the isolate sample, and it is widely suggested for the treatment of *Salmonella* infections [46]. However, we found one stool and one blood isolate resistant to Ceftriaxone from the MIC result, as increasing cases have also been noticed [47,48,49]. The clinical implications of this result need further study. A correlation between bacterial growth rates and antimicrobial treatment in *Salmonella* infections has recently been found [50]. While studies depicted the efficacy of some widely used drugs in the treatment of *Salmonella* infections [51,52], some show an increase in resistance for the same antimicrobial [45,53], leading to consideration of the nature of the clinical isolates strains prior to medication. Nevertheless, in order not to waste time and dispense immediate and adequate treatment upon medical examination, we propose a restriction on antibiotic usage based on the antimicrobial profiles of the endemic prevalent pathogen. The antimicrobial susceptibility profiles and the distribution of virulence genes such as *SpvC* greatly differed from previous molecular epidemiology studies also carried out in Taiwan [43,48,54]. Further investigations are thus needed to attest and comprehend the correlation between *SpvC* and *Salmonella* virulence.

## 4. Materials and Methods

### 4.1. Bacterial Samples and Cell Cultures

A total of 50 *Salmonella* isolates were used in this study. The pathogens were obtained from hospitalized children at National Cheng Kung University Hospital in Tainan, Taiwan. The identified culture sites were categorized as stool and blood. A total of 25 of the samples were recovered from patients’ blood and 25 from patients’ stool. Before considering this bacterial sample size, we initially focused on 12 isolates, six isolated from patients’ blood and six from stool samples, for microbiological investigations (Table 3). These 12 isolates were eventually included in the larger sample of 50 *Salmonella* isolates. Data on patient’s age, *Salmonella* serogroup and culture site were obtained from medical records. All *Salmonella* isolates were cultured and identified according to standard microbiological procedures in the microbiology laboratory of the Department of Pathology [55]. The bacteria were stored at −80 °C in Luria Bertani (LB broth-Miller, AthenaES, Baltimore, MD, USA) media, supplemented with 50% glycerol (Sigma-Aldrich, Darmstadt, Germany). This study was approved by the Institutional Review Board (IRB) of National Cheng Kung University Hospital (NCKUH-IRB approval number: A-ER-103-113, 2018/1/17). Our data were fully de-identified and anonymized to protect the participants’ privacy.

Human fetal small intestine FHs 74 Int (ATCC^®^CCL-241TM, Manassas, VA, USA) cells were grown in a complete growth medium of Hybri-Care Medium ATCC 46-X (ATCC, Manassas, VA, USA) supplemented with 30 ng/mL epidermal growth factor (EGF, Invitrogen, Carlsbad, CA, USA), and 10% fetal bovine serum (Gibco by Life Technologies, Grand Island, NE, USA), and were maintained at 37 °C in a 5% CO_2_ and humidified atmosphere.

### 4.2. Bacterial Adhesion and Invasion Assay

The bacterial adhesion and invasion assays were based on a modified method described by Gagnon et al. [56]. An overnight bacterial culture was 1:100 sub-cultured in fresh LB broth for 2 h at 37 °C under 200 rpm shaking. Following the absorbance measurement at 600 nm, the bacteria were collected using centrifugation at 6000× *g* for 10 min and washed twice with sterile phosphate-buffered saline (PBS, Protech, Taipei, Taiwan). The *Salmonella* isolates inoculum was prepared in complete growth media with a Multiplicity of Infection (M.O.I) equal to 50. Beforehand, FHs 74 Int cells at the late exponential stage were used to seed a 24-well polystyrene plate, followed by an incubation at 37°C for 48 h in a 5% CO_2_ atmosphere, until the cells reached approximately an 80% confluence. After infection, the plates were incubated for 2 h at 37 °C in 5% CO_2_ and washed with PBS to remove non-adherent bacteria.

For the bacterial adhesion test, 200 µL of 0.1% Triton X-100 (AppliChem, Darmstadt, Germany) was added to each well, and the plate was incubated for 10 min at 37 °C to allow cell lysis. Bacterial suspensions were collected and serial 10-fold dilution with 0.85% Saline Solution was achieved. Aliquots of 100 µL bacterial suspension were spread on LB-agar (Hispanagar, Burgos, Spain) plate. The plates were incubated overnight at 37 °C, and the bacterial colonies were counted.

The process followed for the bacterial invasion test was similar to that of bacterial adhesion assay. Subsequent to cell infection, 200 µL of complete growth media containing 150 µg/mL of Gentamicin (Gibco by Life Technologies, Grand Island, NE, USA) was added to each well in order to eliminate the bacteria outside the cells, and the plate was incubated for a further 1 h at 37 °C. Cell lysis, bacterial dilution, and plating were performed as previously described. The *Salmonella* adhesion and invasion rate were evaluated as the percentage ratio of the number of adhered-to or invaded bacteria relative to the initial number of bacteria in the inoculum.

### 4.3. Bacterial Growth Curve

One colony of an overnight *Salmonella* culture streaked onto a LB-agar plate was suspended into 2 mL LB broth and incubated for 15 h at 37 °C under 200 rpm shaking. The bacterial culture was 1:100 diluted into a fresh LB broth or an M9 minimal media and incubated from 0 to 25 h. M9 media contained, at the final concentration, 72 mM Na_2_HPO_4_-2H_2_O, 22 mM KH_2_PO_4_, 8.56 mM NaCl, 18.7 mM NH_4_Cl, 2 mM MgSO_4_, 0.4% glucose, and 0.1mM CaCl_2_. The Optical Density (OD) was measured hourly using a spectrophotometer at a wavelength of 600 nm. A 1:5 dilution of bacteria culture was performed prior to the measurement to keep the values within the accuracy range.

### 4.4. Bacterial Motility

The swarming and swimming motility of *Salmonella* were investigated by pipetting 2 µL of an overnight bacterial culture onto the center of a swarming or swimming plate. The swarm plates were composed of LB and 0.5% agar powder supplemented with 0.5% glucose (D-(+)-Glucose, anhydrous, VWR International, Pennsylvania, USA), and the swim plates were made of LB and 0.3% agar. The plates were prepared a day prior to the experiment. After inoculation, the plates were incubated for 6 h at 37 °C with the lid facing up. Photographs of the plates were taken and processed by using ImageJ software (National Institutes of Health, Bethesda, Maryland, USA) for evaluation of the swarming and swimming area. Isolates that spread on the agar plate away from the initial spot of inoculation were considered motile, whereas non-motile isolates were identified as almost static to where the inoculum was pipetted.

### 4.5. Salmonella Isolates Biofilm Formation

A crystal violet colorimetric assay was used for the evaluation of *Salmonella* biofilm formation based on a conjoined modified method used by Kalai Chelvam et al. [28] and Peng [57]. An overnight bacterial culture was 1:100 diluted into fresh LB and M9 media. Then, 100 µL of the diluted suspension was pipetted into a 96-well polystyrene microtiter plate (JetBiofil, Guangzhou, China) and incubated for 48 h at 37 °C. Non-adherent cells were washed out three times with distilled water, and the plate was dried at room temperature. Staining was achieved by adding 100 µL of 0.4% crystal violet (*v/v*) into each well, followed by incubation in the dark for 20 min. Afterward, the dye was discarded by successive washes with distilled water. Adherent cells were solubilized into 100 µL anhydrous ethanol (Nihon Shiyaku Reagent, Kyoto, Japan), and absorbance was measured at 590 nm. Incubated with the crystal violet dye, adherent cells display a purple color, whose solute optical intensity is proportional to the biofilm biomass produced. The assay was carried out in triplicate with wells containing only media and no bacteria as blank. For data interpretation, three standard deviations above the mean of OD of the negative control (blank) were defined as the cut-off OD, as suggested by Díez-García et al. [58]. Thereby, the isolates were classified into non-biofilm, weak biofilm, moderate biofilm, or strong biofilm producers.

### 4.6. Virulence Genes Genotyping

Multiplex PCR method was performed on the 12 clinical isolates including positive and negative controls. The targeted genes and their corresponding primers are listed in Table 4.

Initially, monoplex PCR was performed, and the products were sequenced to verify primer specificity. Based on the size of the amplicons, the multiplex PCR was optimized by establishing two reaction groups. The first group simultaneously amplified *FliC*, *SpvC*, and *InvA* genes, and the second group *FimH*, *SopB*, and *HilA* genes.

An overnight bacterial cultured on an LB-agar plate was used for each trial. The crude bacterial extract was prepared by suspending three bacterial colonies into 60 µL of lysis buffer (Tris-EDTA, pH 8, 0.1% Triton X-100). The mixture was then boiled at 98 °C for 5 min and spun down at 13,000 g for 10 min. The supernatant was collected and used for further amplification steps.

To obtain a final volume of 50 µL, the multiplex PCR reaction mixture contained 2 µL of the crude bacterial extract, 1 µL of 10 µM forward and reverse primers for each group of genes, 25 µL of One Taq^®^2X MM w/Standard Buffer (New England Biolabs, Massachusetts, USA) and 17 µL autoclaved deionized water.

The reaction was set in a 2720 Thermal Cycler (Applied Biosystems, California, USA) under the following conditions: initial denaturation at 95 °C for 3 min followed by 30 cycles of second denaturation at 94 °C for 30 s, annealing at 58 °C for 30 s, elongation at 72 °C for 2 min, and final extension at 72 °C for 10 min with a hold at 4 °C. The PCR products were electrophoresed on 3% Agarose gel (Invitrogen, Carlsbad, CA, USA ) pre-stained with Health View TM nucleic Acid Stain (Genomics Bioscience and Technology, Taipei, Taiwan). The band pattern was revealed under UV light using the Syngene U: Genius imaging system.

### 4.7. Quantitative-PCR (qPCR) Analysis

In order to provide wider insight into the genetic profile of *Salmonella* isolates, relative quantification of targeted virulence genes was implemented by using the qPCR method with SYBR green reagent on the 50 isolates. An overnight bacterial culture was 1:100 sub-cultured in 5 mL LB broth to an OD between 0.4 and 0.8. Bacteria were collected by centrifugation at 6000× *g* for 10 min and washed twice with PBS. Total RNA was extracted using RNeasy Mini Kit (Qiagen, Hilden, Germany), following the manufacturer’s instructions. RNA concentration was determined using NanoDrop 2000 spectrophotometer (Thermofisher, Massachusetts, USA ) and RNA integrity was evaluated on agarose gel electrophoresis. Complementary DNA (cDNA) was synthesized using iScript cDNA Synthesis Kit (BioRad, California, USA), as described by the manufacturer, and samples were conserved at −20 °C prior to downstream experiments.

The primers used for this step were designed and listed in Table 5. The primers’ annealing temperature was later optimized by gradient PCR. The qPCR test tubes (Applied Biosystems, Massachusetts, USA) contained 10 µL of duplicate sample mix constituted of 5 µL of Power SYBR Green PCR Master Mix (Applied Biosystems), 0.5 µL of Forward and Reverse primer relative to the final concentration of 500 nM, 1 µL of synthesized cDNA, and 3 µL of PCR grade water. No template control tubes were included. StepOnePlus Real-Time PCR System (Applied Biosystems, Massachusetts, USA) was used to perform qPCR runs at the following conditions: 95 °C for 10 min, 40 cycles of 95 °C for 15 s, and 58 °C for 1 min. Amplification specificity was assessed by melting curve. The Ct values that were obtained were normalized against the RNA polymerase sigma factor *RpoD* gene, and the expression of mRNA was calculated using the 2^−∆Ct^, a variation of the Livak method [59,60].

### 4.8. Antibiotic Susceptibility Test and MIC

The *Salmonella* isolates antibiotic susceptibility test was conducted based on the Kirby–Bauer method [61]. The bacterial suspension of the 50 isolates, adjusted to a 0.5 McFarland standard, was spread over Mueller–Hinton Agar (Hi-media, Mumbai, India) using sterile swabs. The experiment was performed using commercial antibiotic discs (Hi-Media, India) of chloramphenicol (C, 30 µg), ciprofloxacin (CIP, 5 µg), ampicillin (AMP, 10 µg), gentamicin (GEN, 10 µg), ceftriaxone (CTR, 30 µg) and trimethoprim-sulfamethoxazole (COT, 1.25/23.7 µg). The antibiotic discs were individually placed on top of the inoculated agar plate and incubated for 18 h at 37°C. Zone diameter sizes were measured, and the results were interpreted according to the Clinical Laboratory Standards Institute (CLSI) and European Committee on Antimicrobial Susceptibility Testing (EUCAST) guidelines, provided by the company along with the antibiotics.

For serotyping and determination of the MIC, all 50 isolates were sent to Ruei Fu Shih Medical Lab at Taichung (Taiwan). By using a standard procedure on the BD Phoenix™ Automated Microbiology System, the MIC of 17 antimicrobial agents including amikacin (AK), ampicillin (AMP), ampicillin–sulbactam (A/S), cefazolin (CZ), cefepime (CPM), cefmetazole (CMZ), cefotaxime (CTX), ceftazidime (CAZ), ceftriaxone (CTR), ertapenem (ETP), gentamicin (GEN), imipenem (ETP), meropenem (MRP), minocycline (MI), piperacillin–tazobactam (PIT), tigecycline (TGC), and trimethoprim–sulfamethoxazole (COT) was resolved. The MIC values were interpreted according to antimicrobial breakpoints in Appendix A.

### 4.9. Statistical Analysis

The data were analyzed using SPSS statistics 17.0 software (IBM Corporation, New York, USA). One-way or two-way analysis of variance (ANOVA) pairwise comparison with Bonferroni corrections was made to evaluate significant differences. The results were significantly different when the *p* value was less than or equal to 0.05.

## 5. Conclusions

Our data ultimately show that selected clinical isolates have various virulence factor profiles, which may or not be correlated. The pathogenic determinants contributing to the clinical symptoms and the persistence of the pathogens within the host might be complex and underestimated. Thus, these factors could not be entirely covered by the present study. Almost all 50 isolates expressed *InvA*, *FimH*, *SopB*, and *HilA* mRNA, but the expression of *FliC* and *SpvC* was highly heterogeneous. Notwithstanding some limitations in this study, we believe that the presence of *SpvC* might paradoxically attenuates the virulence of the *Salmonella* isolates. For future consideration, the generation and testing of isogenic mutants and their complemented variants could help to clear up this hypothesis.

## Figures and Tables

**Figure 1 pathogens-10-00074-f001:**
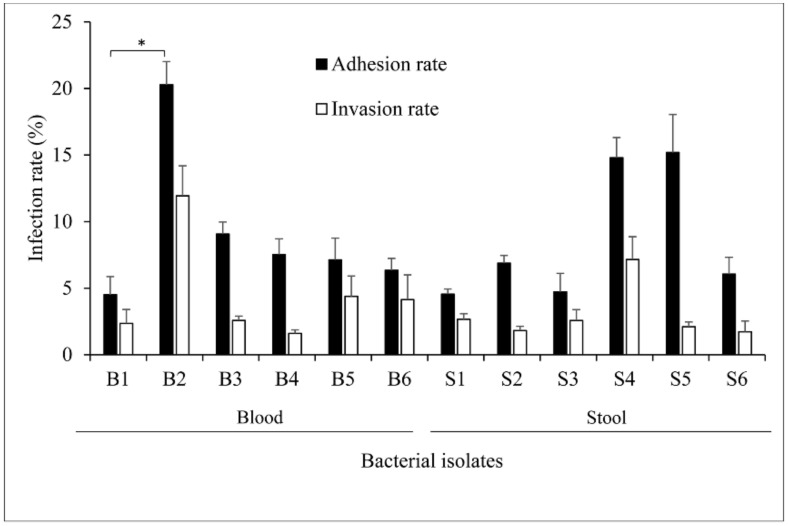
Infection rates of different *Salmonella* isolates on FHs 74 Int cells. Bacteria inoculum suspended in complete Hybri-Care Medium was directly added to the cultured cells in a 24-well plate to a multiplicity of infection (M.O.I_ of 50:1 and incubated for 2 h at standard conditions. Bacterial colony forming unit (CFU) count was achieved after cell wash and lysis. The adhesion rate was calculated as the percentage ratio of bacteria attaching the cell membrane and probably penetrating the cells to the initial concentration of bacteria in the inoculum. The invasion rate solely considered the engulfed bacteria since Gentamicin treatment discarded the external bacteria. Values were expressed as the mean of four independent repeats with standard error of the mean (±SEM). A general linear model including repeated measures was used for statistical analysis. (*) *p* = 0.007.

**Figure 2 pathogens-10-00074-f002:**
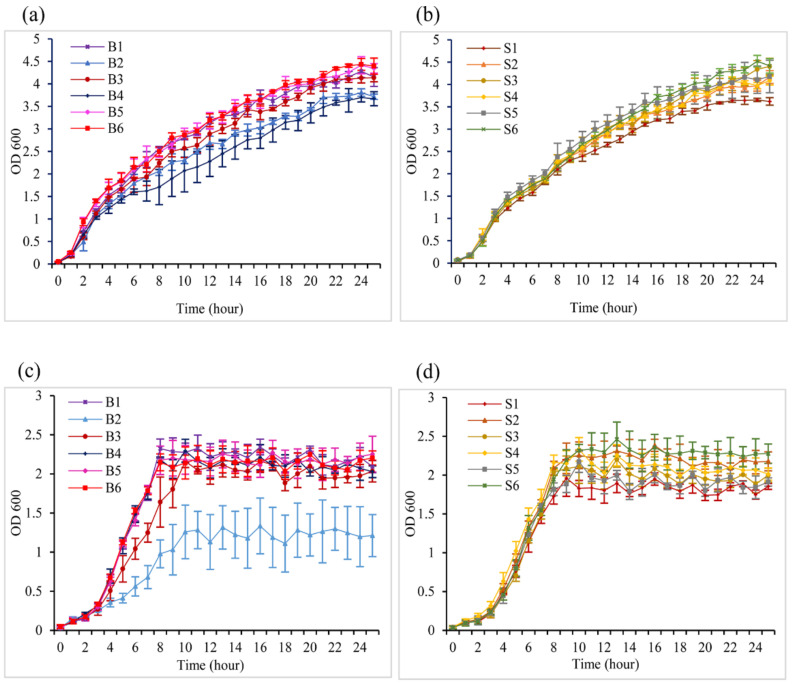
Growth curves of different *Salmonella* isolates. (**a**) Blood isolates in LB medium. (**b**) Stool isolates in LB. (**c**) Blood isolates in M9 medium. (**d**) Stool isolates in M9 medium. The optical density (OD) of cultured bacteria was read at 600 nm. The values represented the mean of three independent measurements with the standard error of the mean (±SEM).

**Figure 3 pathogens-10-00074-f003:**
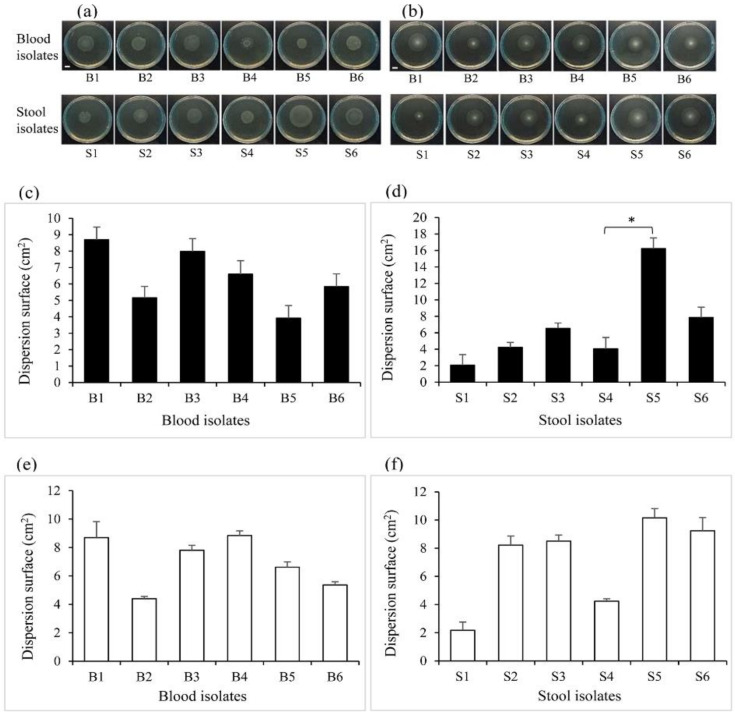
The motility patterns of the *Salmonella* isolates. *Salmonella* isolates (**a**) swarming and (**b**) swimming pattern revealed on agar plates. The pictures were taken 6 h after plate inoculation. Swarming dispersion surface of (**c**) blood and (**d**) stool isolates on 0.5% agar supplemented with 0.5% glucose. Swimming dispersion surface of (**e**) blood and (**f**) stool isolates on 0.3% agar plates. Comparative dispersion was represented as the mean of three repeats with the standard error of the mean (±SEM). One-way analysis of variance was used for statistical analysis. (*) *p* = 0.03. The scale bar corresponds to 1 cm.

**Figure 4 pathogens-10-00074-f004:**
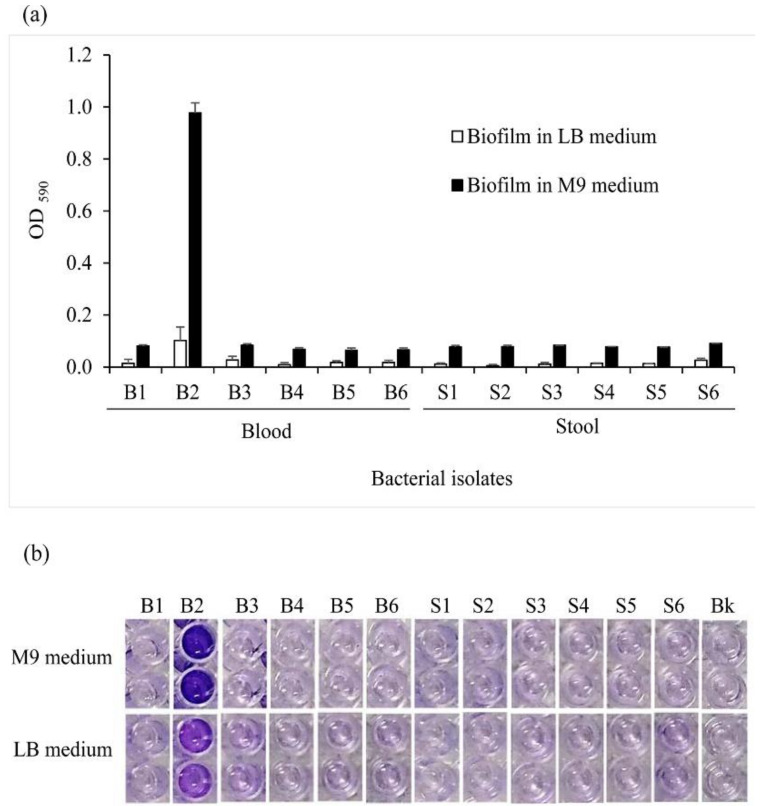
Biofilm formed by *Salmonella* isolates. (**a**) Comparative optical density (OD) of bacterial forming biofilm cultured in LB and M9 medium. Bacteria adhering to the well of a 96-well plate after an incubation of 48 h at 37 °C were stained with crystal violet and dissolved into anhydrous ethanol. The OD was eventually measured at 590 nm with consideration of a negative control referred to as blank. By using the blank cut-off OD (ODc), isolates have been identified as non-biofilm, weak, or strong biofilm-forming when OD ≤ ODc, ODc < OD ≤ (2 × ODc), and (4 × ODc) < OD, respectively. Two-way analysis of variance method was used for statistical analysis. (**b**) Visual representation of biofilm formed by *Salmonella* isolates stained and dissolved in an ethanolic solvent. Data presented as the mean ± SEM of three measurements performed in three independent experiments. Bk: blank.

**Figure 5 pathogens-10-00074-f005:**
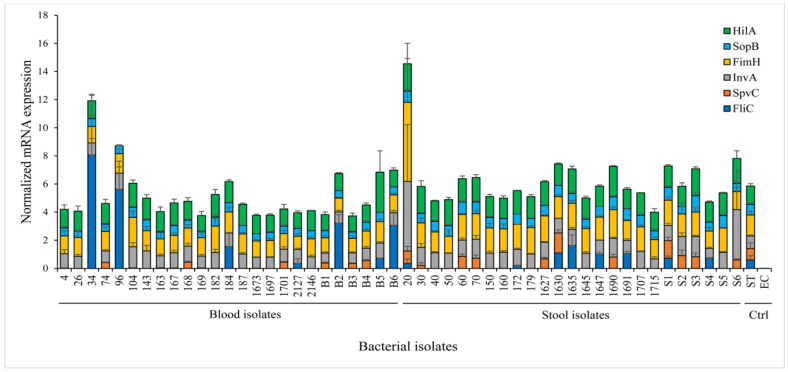
Quantitative mRNA expression of virulence genes in *Salmonella* isolates. Total RNA from bacteria cultured in LB was extracted. The cDNA was synthesized and used for qPCR. The sample Ct values were first normalized against the endogenous *RpoD* gene control, then the level of gene expression was calculated by using the formula 2 ^−∆Ct^. Data were presented as mean ±SEM of three independent experimental repeats. *Salmonella* Typhimurium LT2 (ST, positive control), and *E. coli* (EC, negative control). Ctrl: control.

**Table 1 pathogens-10-00074-t001:** Distribution of virulence genes among the bacterial isolates.

Bacterial Isolates	*FliC*	*SpvC*	*InvA*	*FimH*	*SopB*	*HilA*
B1	-	+	+	+	+	+
B2	+	-	+	+	+	+
B3	-	+	+	+	+	+
B4	-	+	+	+	+	+
B5	+	-	+	+	+	+
B6	+	-	+	+	+	+
S1	+	+	+	+	+	+
S2	-	+	+	+	+	+
S3	-	+	+	+	+	+
S4	+	-	+	+	+	+
S5	-	-	+	+	-	+
S6	-	+	+	+	+	+

+: Presence of gene, -: Absence of gene.

**Table 2 pathogens-10-00074-t002:** *Salmonella* isolates and antibiotic susceptibility profiles.

Antimicrobials	Susceptibility Characteristics (*n* = 50)		
	Sensitive N (%)	Intermediate N (%)	Resistant N (%)
Chloramphenicol	32 (64%)	0 (0%)	18 (36%)
Ciprofloxacin	2 (4%)	38 (76%)	10 (20%)
Ampicillin	24 (48%)	1 (2%)	25 (50%)
Gentamicin	22 (44%)	24 (48%)	4 (8%)
Ceftriaxone	45 (90%)	4 (8%)	1 (2%)
Trimethoprim-Sulfamethoxazole	36 (72%)	0 (0%)	14 (28%)

N: number; %: percentage.

**Table 3 pathogens-10-00074-t003:** *Salmonella* enterica used for microbiology study.

Bacterial Identification Name	Bacterial Collection Number	Isolation Source	Serotyping Group
B1	1517	Blood	D1
B2	1596	Blood	B
B3	1462	Blood	D1
B4	1793	Blood	D1
B5	1829	Blood	B
B6	1833	Blood	B
S1	1792	Stool	B
S2	1785	Stool	D1
S3	1794	Stool	D1
S4	1795	Stool	B
S5	1798	Stool	B
S6	1762	Stool	D1

**Table 4 pathogens-10-00074-t004:** Targeted *Salmonella* virulence genes and their primers.

Target Gene	Oligo Sequence	Amplicon Size (bp)	NCBI Reference Sequence
*FliC*	F: AACGCAGTAAAGAGAGGACG	162	NC 003197.2
	R: ACACCGTAAACAACCTGACT		
*SpvC*	F: TCACGTAAAGCCTGTCTCTG	214	NC 003277.2
	R: CAGACCAGGAAAATTCGCAG		
*FimH*	F: CGGAAAAATCAGGTTGGGTC	360	NC 003197.2
	R: CGGTAGAGGTCGTCACATAG		
*SopB*	F: CTTTTGCAGGTAAGCCATCC	235	NC 003197.2
	R: CACTCGCTGCATAACCTCTA		
*InvA*	F: TCCACGAATATGCTCCACAA	381	NC 003197.2
	R: CCAACAATCCATCAGCAAGG		
*HilA*	F: GCTTTAGGATTACTGGGGCT	407	NC003197.2
	R: AGTCCTGTTATTTCCTGCGT		

F: Forward; R: Reverse.

**Table 5 pathogens-10-00074-t005:** Primers implemented for qPCR.

Target Gene	Oligo Sequence
*FliC*	F: TCTGTACCGCACCCAGGTCA
	R: CGGCTGCTACAACCACCGAA
*SpvC*	F: CACTCTGGCGCATCCCTGAA
	R: GCTGCTTATGATGGGGCGGA
*FimH*	F: TGACGATCCCTCGCCAGACA
	R: ACGACCTGTCCGGCATTCAC
*SopB*	F: GTATTGCGCCAGCCATTCGC
	R: GCCGAGGCGCTACATCAGTT
*InvA*	F: CGACTTCCGCGACACGTTCT
	R: TAGCCTGGCGGTGGGTTTTG
*HilA*	F: TCGTAGTGGTGTCTCCGCCA
	R: CTCAGGCCAAAGGGCGCATA
*RpoD*	F: GTTGACCCGGGAAGGCGAAA
	R: GGGTATTCGGCAACGGAGCA

F: Forward; R: Reverse.

## Data Availability

Data is contained within the article or Appendix A.

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
