# Peer review of "Correlation between Pathogenic Determinants Associated with Clinically Isolated Non-Typhoidal Salmonella"

_pathogens, 2021, doi:10.3390/pathogens10010074_

Round 1

Reviewer 1 Report

Manuscript ID: pathogens-1056170

Title: Correlation between pathogenic determinants associated with clinically isolated non-typhoidal Salmonella

Comments to the Author

In their manuscript the authors present new and interesting data. Overall, this is a generally well written that has used sound and relevant methodologies. However, I have one major concern that the authors may wish to consider.

Major concern:

Line 79-81: The authors stated “Herein, we seek to investigate the distribution of virulence traits among Salmonella clinical isolates and comprehend the conditions sustaining bacteria invasiveness in vitro” as opposed to the title “Correlation between pathogenic determinants 2 associated with clinically isolated non-typhoidal 3 Salmonella”. It is not clear how many isolates were subjected for virulence genotyping. Line 222 – 241 shows only 12 isolates (out of 50) were subjected to virulence genes genotyping [Line 465: 4.6. Virulence genes genotyping]. What is the basis to select 12 isolates for genotyping and why? How do you relate Figure 5 to Table 1? Regardless, in their conclusion stated that “Almost all the 50 isolates expressed InvA, FimH, SopB, and HilA virulence genes but the expression profile of FliC and SpvC was highly variable [Line 539-541].” This is confusing, and please clarify it.

Minor correction: It is difficult to read Table 3 drug abbreviations in three letters. Please re-arrange the Table 3 into ‘landscape’.  

Author Response

Reviewer 1

Observation-Major concern

Line 79-81: The authors stated “Herein, we seek to investigate the distribution of virulence traits among Salmonella clinical isolates and comprehend the conditions sustaining bacteria invasiveness in vitro” as opposed to the title “Correlation between pathogenic determinants 2 associated with clinically isolated non-typhoidal 3 Salmonella”. It is not clear how many isolates were subjected for virulence genotyping. Line 222 – 241 shows only 12 isolates (out of 50) were subjected to virulence genes genotyping [Line 465: 4.6. Virulence genes genotyping]. What is the basis to select 12 isolates for genotyping and why? How do you relate Figure 5 to Table 1? Regardless, in their conclusion stated that “Almost all the 50 isolates expressed InvA, FimH, SopB, and HilA virulence genes but the expression profile of FliC and SpvC was highly variable [Line 539-541].” This is confusing, and please clarify it.”

Responses:

We highly appreciate the reviewer’s insightful and helpful comments. We address here in successive order the highlighted concerns.

  • The statement at the end of the Introduction section has been carefully rewritten to reinforce the logic of the work. The change can be found at lines 80 to 83 of the tracked manuscript: “Herein, we seek to examine the correlation between virulence determinants in association with Salmonella pathogenicity by investigating the distribution of virulence traits among Salmonella clinical isolates and by comprehending their interconnection.”

  • We used two different methods to investigate targeted virulence genes in Salmonella The first method firstly employed was the genotyping by Multiplex PCR which was realized on 12 isolates and that to also keep a logical schema following the series of microbiological analysis considered (adhesion/invasion, growth rate, motility, and biofilm assays). Then, we further realized the qPCR to quantitatively detect the expression of the same virulence genes in the 50 isolates. To clear up this misunderstanding, amendments have been made at line 223 “2.5. Salmonella isolates genotyping by multiplex PCR”, line 463 “Multiplex PCR method was performed on the 12 clinical isolates including positive and negative controls”, lines 295-300 of the Discussion, and lines 385-388 of the Materials and Methods.

Initially, a qPCR assay was performed with the 12 isolates and we found that the result, regarding the expression of targeted genes, perfectly confirmed that of the genes’ detection via multiplex PCR. Performing the multiplex PCR for the 50 isolates was then judged to produce redundant results since the qPCR was also selected as an effective method to detect and quantify virulence genes (Source: A Basic Guide to Real Time PCR in Microbial Diagnostics: Definitions, Parameters, and Everything, Kralik P. and Ricchi M., Front. Microbiol., 2017). This explain why only 12 isolates were selected for multiplex PCR instead of all the 50 isolates.

Moreover, by considering that qPCR indicates the presence of specific gene via the quantification of mRNA, we compared the amount of mRNA expressed between the isolates and the positive control S. Typhimurium. The isolates that do not substantially express mRNA for the targeted virulence gene were then identified to lack this gene. This allowed us to correlate the results between Figure 5 and Table 1.

  • We have then updated the conclusion of our manuscript in order to clarify the confusion that has been brought to our attention. The modifications are presented at lines 537-538 “Almost all the 50 isolates expressed InvA, FimH, SopB, and HilA mRNA but the expression of FliC and SpvC was highly heterogeneous.”

Observation-Minor correction

It is difficult to read Table 3 drug abbreviations in three letters. Please re-arrange the Table 3 into ‘landscape’.”

Response

Thank you for the suggestion. We have now rearranged the Table 3 (now Table S1) into “landscape”.

Reviewer 2 Report

The manuscript entitled: Correlation between pathogenic determinants associated with clinically isolated non-typhoidal Salmonella, characterizes strains isolated from blood and faeces from children in Taiwan.

The test methods were well selected and well described. However, no positive control was used in the study of adhesion and invasiveness, and in the M9 and LB growth assays. This is important because the signals that trigger SPI-1 induction and mobility are not well understood, and various growth conditions are used in the field of Salmonella research to identify invasive bacteria.

Regarding the tested strains, it is unclear why, of the 50 strains selected for the study, only 12 are widely analysed. It must be clearly justified.

The description of the results is very synthetic and therefore not fully clear as in paragraph 2.4. The effects of the intrinsic nature of the bacteria and culture medium on biofilm formation. Where does the purple colour come from? A little description of the method is needed for clarification.
Table 3. Antimicrobial susceptibility summary from MIC test - should be included in the supplement.

In the discussion, the authors try to interpret the obtained results by introducing hypotheses that were neither intended for the purpose of the research nor in the content of the introduction.
In addition, line 302-303 and line 320-321- all isolates…….. - while only 12 were characterized out of 50. The discussion should be rewritten and / or the introduction modified.

Author Response

Reviewer 2

Observations:

“The test methods were well selected and well described. However, no positive control was used in the study of adhesion and invasiveness, and in the M9 and LB growth assays. This is important because the signals that trigger SPI-1 induction and mobility are not well understood, and various growth conditions are used in the field of Salmonella research to identify invasive bacteria.

Regarding the tested strains, it is unclear why, of the 50 strains selected for the study, only 12 are widely analysed. It must be clearly justified.

The description of the results is very synthetic and therefore not fully clear as in paragraph 2.4. The effects of the intrinsic nature of the bacteria and culture medium on biofilm formation. Where does the purple colour come from? A little description of the method is needed for clarification.

Table 3. Antimicrobial susceptibility summary from MIC test - should be included in the supplement.

In the discussion, the authors try to interpret the obtained results by introducing hypotheses that were neither intended for the purpose of the research nor in the content of the introduction.

In addition, line 302-303 and line 320-321- all isolates…….. - while only 12 were characterized out of 50. The discussion should be rewritten and / or the introduction modified.”

Responses:

We highly appreciate the reviewer’s insightful and helpful comments. We will address each point accordingly.

  • We perfectly agree with the reviewer for the importance of the positive controls. However, as we mentioned in the discussion, we were concerned about the overgeneralization of our present results. During the collection of data, several hypotheses have been raised displaying the wideness of the present topic whose different points cannot be extensively covered in the present manuscript. As part of our future research, we will investigate in depth the association between SPI-1 which mediates contact-dependent invasion of host cells and bacterial motility as well as the environmental and growth conditions driving bacterial invasiveness. We thank you very much for this suggestion.

  • To improve the clarify of why 12 isolates out of the 50 were widely analyzed, we introduced explanatory sentences at the discussion lines 295 to 300 “Initially, microbiological analysis was conducted on 12 Salmonella clinical isolates which included adherence and invasion assay, growth, motility, biofilm formation, and genotyping assays. For consideration of a more representative sample size and for wider bacterial screening, the number of the isolates was eventually increased to a total of 50. The 50 isolates were then considered for the quantification of virulence gene expressions and antimicrobial testing.” Moreover, sentences in the first paragraph of the Materials and Methods section 1. Bacterial samples and cell cultures (lines 385 to 388) were modified.

  • The purple color comes as a result of adherent bacteria cells stained by the crystal violet dye. As suggested, this detail has been included to the appropriate method section at lines 455 to 457 “Incubated with the crystal violet dye, adherent cells display a purple color which solute optical intensity is proportional to the biofilm biomass produced.”

  • The Table 3. Antimicrobial susceptibility summary from MIC test has be moved to the Supplementary Materials and renamed Table S1. Therefore, the supplementary materials in text and at the section after the conclusions (lines 544-546) of the manuscript have been revised.

  • The discussion section has been updated and the changes are highlighted using the “Track changes” function. Clarifications have been made as suggested and sentences have been carefully re-written. To improve the logic flow of the manuscript and clear up the confusion noticed by the reviewers, we have also re-organized the paragraphs 6.Comparative virulence gene expression and serotyping and 2.7. Salmonella isolates in vitro susceptibility to antibiotics at the Results section, the lines 293 to 295 and 347 to 378 of the Discussion, and the paragraphs 4.7. Quantitative-PCR (qPCR) analysis and 4.8. Antibiotic susceptibility test and MIC of the Materials and Methods part. The English language and style of the manuscript were also thoroughly improved.

Round 2

Reviewer 2 Report

The authors introduced correction according to the comments.